# Therapeutics Data Commons: Machine Learning Datasets and Tasks for Drug Discovery and Development

**Kexin Huang**[1]*  **Tianfan Fu**[2]*  **Wenhao Gao**[3]*  **Yue Zhao**[4], **Yusuf Roohani**[5],
**Jure Leskovec**[5], **Connor W. Coley**[3], **Cao Xiao**[6], **Jimeng Sun**[7], **Marinka Zitnik**[1]
[1]Harvard [2]Georgia Tech [3]MIT [4]CMU [5]Stanford [6]Amplitude [7]UIUC
contact@tdcommons.ai

## Abstract

Therapeutics machine learning is an emerging field with incredible opportunities for innovation and impact. However, advancement in this field requires formulation of meaningful tasks and careful curation of datasets. Here, we introduce Therapeutics Data Commons (TDC), the first unifying platform to systematically access and evaluate machine learning across the entire range of therapeutics. To date, TDC includes 66 AI-ready datasets spread across 22 learning tasks and spanning the discovery and development of safe and effective medicines. TDC also provides an ecosystem of tools and community resources, including 33 data functions and diverse types of data splits, 23 strategies for systematic model evaluation, 17 molecule generation oracles, and 29 public leaderboards. All resources are integrated and accessible via an open Python library. We carry out extensive experiments on selected datasets, demonstrating that even the strongest algorithms fall short of solving key therapeutics challenges, including distributional shifts, multi-scale and multi-modal learning, and robust generalization to novel data points. We envision that TDC can facilitate algorithmic advances and considerably accelerate machine-learning model development, validation and transition into biomedical and clinical implementation. TDC is available at https://tdcommons.ai.

## 1   Introduction

The overarching goal of scientific research is to find ways to cure, prevent, and manage all diseases. With the proliferation of high-throughput biotechnological techniques [65] and advances in the digitization of health information [2], machine learning provides a promising approach to expedite the discovery and development of safe and effective treatments. Getting a drug to market currently takes 13-15 years and between US\$2 billion and \$3 billion on average, and the costs are going up [113]. Further, the number of drugs approved every year per dollar spent on development has remained flat or decreased for most of the past decade [113, 104]. Faced with skyrocketing costs for developing new drugs and long, expensive processes with a high risk of failure, researchers are looking at ways to accelerate all aspects of drug development. Machine learning has already proved useful in the search of antibiotics [136], polypharmacy [176], drug repurposing for emerging diseases [47], protein folding and design [64, 41], and biomolecular interactions [177, 3, 55, 39].

Despite the initial success, the attention of the machine learning scientists to therapeutics remains relatively limited, compared to areas such as natural language processing and computer vision, even though therapeutics offer many hard algorithmic problems and applications of immense impact. We posit that is due to the following key challenges: (1) The lack of AI-ready datasets and standardized knowledge representations prevent scientists from formulating relevant therapeutic questions as

---

*Equal contribution.

35th Conference on Neural Information Processing Systems (NeurIPS 2021) Track on Datasets and Benchmarks.

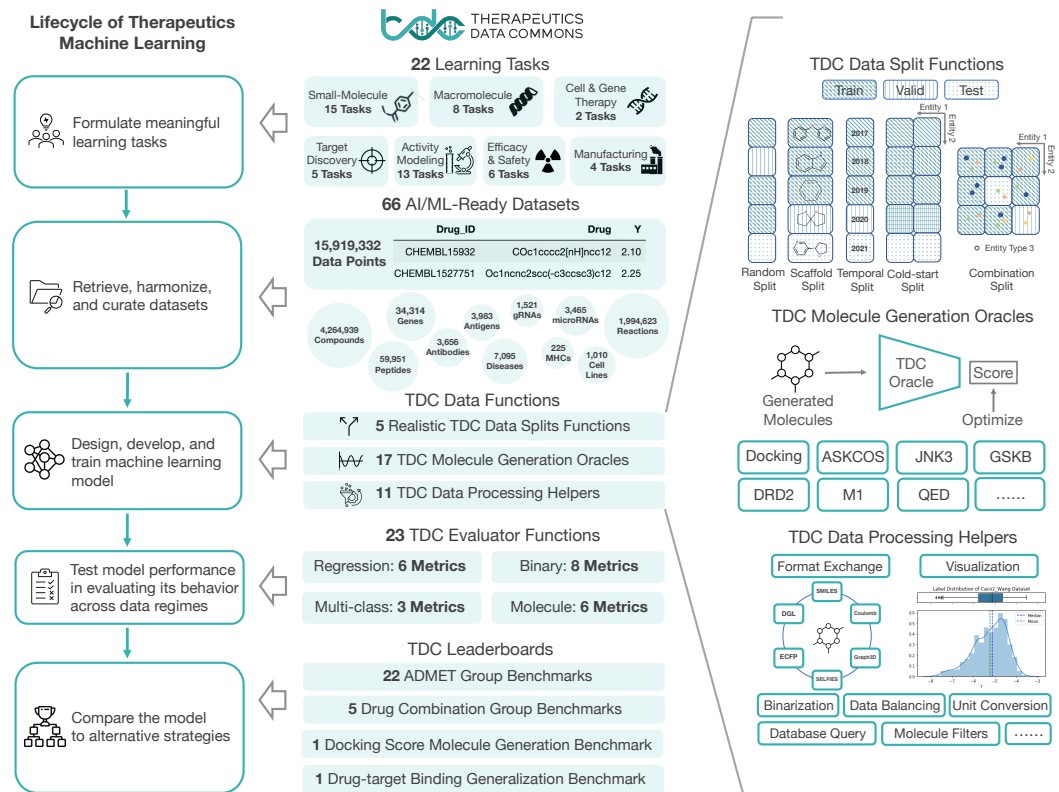

Figure 1: **Overview of Therapeutics Data Commons (TDC).** TDC is a platform with AI-ready datasets and tasks for therapeutics, spanning the discovery and development of medicines. TDC provides an ecosystem of tools and data functions, including strategies for systematic model evaluation, meaningful data splits, data processors, and molecule generation oracles. All resources are integrated and accessible via a Python package. TDC also provides community resources with extensive documentation and tutorials, and leaderboards for systematic model comparison and evaluation.

solvable machine-learning tasks—the challenge is how to computationally operationalize these data to make them amenable to learning; (2) Datasets are of many different types, including experimental readouts, curated annotations and metadata, and are scattered around the biorepositories—the challenge for non-domain experts is how to identify, process, and curate datasets relevant to a task of interest; and (3) Despite promising performance of models, their use in practice, such as for rare diseases and novel drugs in development, is hindered—the challenge is how to assess algorithmic advances in a manner that allows for robust and fair model comparison and represents what one would expect in a real-world deployment or clinical implementation.

**Present work.** To address the above challenges, we introduce Therapeutics Data Commons (TDC), the first platform to systematically access and evaluate machine learning across the entire range of therapeutics (Figure 1). TDC provides AI-ready datasets and learning tasks, together with an ecosystem of tools, libraries, leaderboards, and community resources. To date, TDC contains 66 datasets (Table 1) spread across 22 learning tasks, 23 strategies for systematic model evaluation and comparison, 17 molecule generation oracles, and 33 data processors, including 5 types of data splits. Datasets in TDC are diverse and cover a range of therapeutic products (*e.g.*, small molecule, biologics, and gene editing) across the entire range of drug development (*i.e.*, target identification, hit discovery, lead optimization, and manufacturing). We develop a Python package that implements all functionality and can efficiently retrieve any TDC dataset. Finally, TDC has 29 leaderboards, each with carefully designed train, validation, and test splits to provide a systematic model comparison and evaluation framework and test the extent to which model performance reflects real-world settings.

Datasets and tasks in TDC are challenging. To this end, we rigorously evaluate 21 domain-specific and state-of-the-art methods across 24 TDC benchmarks (Section 4): (1) a group of 22 ADMET benchmarks are designed to predict properties of small molecules—it is a graph representation learning problem; (2) the DTI-DG benchmark is designed to predict drug-target binding affinity using a patent temporal split—it is a domain generalization problem; (3) the docking benchmark is designed

to generate novel molecules with high docking scores in limited resources—it is a low-resource generative modeling problem. We find that theoretic domain-specific methods often have better or comparable performance with state-of-the-art models, indicating urgent need for rigorous model evaluation and an ample opportunity for algorithmic innovation.

Finally, datasets and benchmarks in TDC lend themselves to the study of the following open questions in machine learning and can serve as a testbed for new algorithms, including:

- **Few-shot learning and extrapolation**: Prevailing methods require abundant label information. However, labeled examples are scarce in drug development and discovery, considerably limiting the methods' use for problems that require reasoning about new phenomena, such as novel drugs in development, emerging pathogens, and therapies for rare diseases.
- **Multi-modal and knowledge graph reasoning**: Data points in TDC have diverse representations and are given in various modalities, including graphs, tensors/grids, sequences, and spatio-temporal entities.
- **Distribution shifts**: Candidate drugs and target proteins can quickly change their behavior depending on biological context, such as cellular, tissue, and disease states, meaning that models need to accommodate the underlying distribution shifts and have robust generalizable performance on previously unseen data points.
- **Causal inference**: TDC contains datasets that quantify drug response, the response of molecules and cells to different kinds of perturbations, such as treatment, CRISPR gene over-expression, and knockdown perturbations. Observing how and when a cellular, molecular or patient response is altered can provide clues into underlying mechanisms of the perturbation and, ultimately, disease. Thus, these datasets represent a natural testbed for causal inference.

## 2 Related Work

TDC is the first unifying platform of datasets and learning tasks for drug discovery and development. We briefly review how TDC relates to data collections, benchmarks, and toolboxes in other areas.

**Relation to biomedical and chemical data repositories.** There is a myriad of databases with therapeutically relevant information. For example, BindingDB [89] curates binding affinity data, ChEMBL [98] curates bioassay data, THPdb [148] and TTD [156] record information on therapeutic targets, and BioSNAP Datasets [178] contains biological networks. DrugBank [160] provides rich information around drug products. While these biorepositories are important for data deposition and re-use, they do not contain AI-ready datasets (*e.g.*, well-annotated metadata, requisite sample size, and granularity, provenance, multimodal data dynamics, and curation needs), meaning that extensive domain expertise is needed to process the them and construct datasets that can be used for machine learning. In addition, while each of the above database focus on a single-modality resource, TDC has a wider coverage in extending to emerging therapeutic types such as CRISPR and therapeutics pipelines such as manufacturing.

**Relation to ML benchmarks.** Benchmarks have a critical role in facilitating progress in machine learning (*e.g.*, ImageNet [33], Open Graph Benchmark [52], SuperGLUE [153]). More related to us, MoleculeNet [161] provides datasets for molecular modeling and TAPE [117] provides five tasks for protein transfer learning. In contrast, TDC broadly covers modalities relevant to therapeutics, including compounds, proteins, biomolecular interactions, genomic sequences, disease taxonomies, regulatory and clinical datasets. Further, while MoleculeNet and TAPE aim to advance representation learning for compounds and proteins, TDC has a focus on drug discovery and development.

**Relation to therapeutics ML tools.** Many open-science tools exist for biomedical machine learning. Notably, DeepChem [116] implements models for molecular machine learning; DeepPurpose [54] is a framework for compound and protein modeling; OpenChem [72] and ChemML [48] also provide models for drug discovery tasks. In contrast, TDC is not a model-driven framework; instead, it provides datasets and formulates learning tasks. Further, TDC provides an extensive ecosystem of tools and resources (Section E) for model development and evaluation.

# 3 Overview of Therapeutics Data Commons

TDC has three major components: a collection of datasets and formulations of meaningful learning tasks; a comprehensive ecosystem of tools and community resources to support data processing, model development and validation; and a collection of leaderboards to support fair model comparison and benchmarking. The programmatic access is provided through the TDC Python package[2] (Figure 3). We proceed with a brief overview of each TDC's component.

**1) AI-ready datasets and learning tasks.** TDC has an unique three-tiered hierarchical structure, which to our knowledge, is the first attempt at systematically organizing ML for therapeutics. We organize TDC into three distinct *problems*. For each problem, we provide a collection *learning tasks*, and for each task, we provide a collection of *datasets*.

In the first tier, after observing a large set of therapeutics tasks, we identify three major problems:

- **Single-instance prediction** `single_pred`: Predictions about individual biomedical entities.
- **Multi-instance prediction** `multi_pred`: Predictions about multiple biomedical entities.
- **Generation** `generation`: Generation of biomedical entities with desirable properties.

In the second tier, TDC is organized into tasks. TDC currently includes 22 tasks, covering a range of development pipelines and therapeutic modalities. These range from small molecules to biologics, including antibodies, peptides, microRNAs, and gene therapy. Further, TDC tasks map to the following development pipelines:

- **Target discovery**: Tasks to identify candidate drug targets.
- **Activity modeling**: Tasks to screen and generate individual or combinatorial candidates with high binding activity towards targets.
- **Efficacy and safety**: Tasks to optimize therapeutic signatures indicative of safety and efficacy.
- **Manufacturing**: Tasks to synthesize safe and effective drug.

In the third tier, TDC provides multiple datasets for each task. To date, TDC includes 66 datasets (Table 1). For each dataset, we provide several dataset splits into training, validation, and test sets. TDC datasets vary in size between 200 and 2 million data points. All datasets are harmonized and contain metadata, provenance information, and curated annotations (Appendix B-D).

Notably, all datasets included in TDC are carefully processed from the primary data resources. The raw data come in various file formats, including machine non-readable formats, and is often inaccessible to the users. For each dataset, the raw data can be of different types, including experimental readouts, curated annotations, and metadata, and are scattered around the biorepositories and paper supplementary documents, thus requiring extensive curation to transform/link it to a format that is amenable to ML analyses. Further, many transformations and quality control steps require domain-specific expertise and familiarity with many bioinformatics and cheminformatics tools.

**2) Ecosystem of tools and community resources.** TDC includes numerous data functions that can be readily used with any TDC dataset. TDC divides its programmatic ecosystem into four broad categories (Figure 1) that we describe in detail in Appendix E:

- **23 strategies for model evaluation**: TDC implements a series of metrics and performance functions to debug models, evaluate model performance for any task in TDC, and assess whether model predictions generalize to out-of-distribution datasets.
- **5 types of dataset splits**: TDC implements data splits that reflect real-world settings, including random split, scaffold split, cold-start split, temporal split, and combination split.
- **17 molecule generation oracles**: Molecular design tasks require oracle functions to measure the quality of generated entities. TDC implements 17 molecule generation oracles, representing the most comprehensive collection of oracles, each tailored to measure a distinct quality of generated molecules.
- **11 data processing functions**: Datasets cover a range of modalities, each requiring distinct data processing and quality checks. TDC provides functions for data format conversion,

---

[2]Documentation of TDC Python package can be found at http://tdc.readthedocs.io.

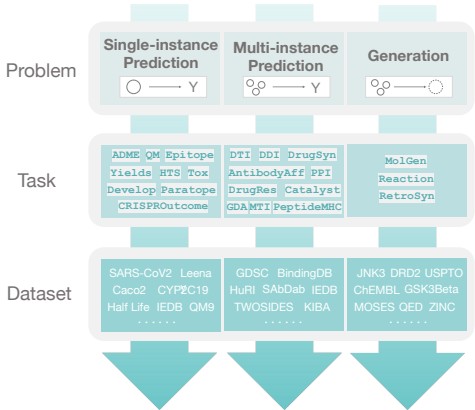

Figure 2: **Tiered design of Therapeutics Data Commons.** We organize TDC into three distinct problems. For each problem, we give a collection of learning tasks. Finally, for each task, we provide a collection of datasets (Section 3). For example, TDC.Caco2_Wang is a dataset under the ADME learning task, which, in turn, is under the single-instance prediction problem. This unique three-tiered hierarchical structure is, to the best of our knowledge, the first attempt at systematically organizing therapeutics ML.

```python
from tdc.single_pred import Tox
data = Tox(name = 'DILI')
split = data.get_split(method = 'random', seed = 42, frac = [0.7, 0.1, 0.2])

from tdc import Evaluator
evaluator = Evaluator(name = 'MSE')
score = evaluator(y_true, y_pred)

from tdc import Oracle
oracle = Oracle(name = 'JNK3')
oracle(['C[C@@H]1CCN(C(=O)CCCc2ccccc2)C[C@@H]1O'])

from tdc import BenchmarkGroup
group = BenchmarkGroup(name = 'ADMET_Group')
predictions = {}

for benchmark in group:
    name = benchmark['name']
    train_val, test = benchmark['train_val'], benchmark['test']
    ## --- train your model --- ##
    predictions[name] = y_pred

group.evaluate(predictions)
```

Figure 3: **TDC Python package** (PyTDC). All resources in TDC, including data loaders, data split functions, molecule generation oracles, data processing helpers, and model evaluators (Figure 1) can be easily accessed via our Python package. The installation of the TDC package is hassle-free (*e.g.*, using PyPI package management system) with minimum dependency on external packages. In this example, we first create a DataLoader object and use it to obtain a random split of the TDC.DILI dataset. The second and third code blocks illustrate how to access TDC data functions, *i.e.*, an MSE model evaluator and a JNK3 molecule generation oracle. Lastly, a BenchmarkGroup object provides support for TDC leaderboards. See also Appendix F, and documentation and tutorials on Github and TDC website.

> visualization, binarization, data balancing, unit conversion, database querying, molecule filtering, and more.

**3) Leaderboards.** TDC provides leaderboards for systematic model evaluation and comparison. For a model to be useful for a particular therapeutic question, it need to perform well consistently across multiple related datasets and tasks. For this reason, we group individual benchmarks in TDC into meaningful groups, which we refer to as *benchmark groups*. Datasets and tasks within a benchmark group are carefully selected and centered around a particular therapeutic question. Further, dataset splits and evaluation metrics are carefully selected to reflect the real-world requirement. The current release of TDC has 29 leaderboards (29 = 22 + 5 + 1 + 1; see Figure 1). Section 4 describes 24 of them and reports extensive empirical results for them. We follow the mechanisms based on previous successes [52, 70], where the test set label is public and users are required to explicitly provide consent to an honor code and open-source their models with fully reproducible codes.

## 4 Experiments on Selected Datasets

TDC benchmarks and leaderboards enable systematic model development and evaluation. We illustrate them through three examples. Datasets, code, and evaluation strategies for these experiments are available at https://github.com/mims-harvard/TDC/tree/master/examples.

### 4.1 Twenty-Two Datasets in the ADMET Benchmark Group

**Motivation.** Although millions of active compounds have been identified, the number of approved new drugs has not drastically increased in recent years [104]. Besides the non-technical issues, the efficacy and safety deficiencies are main factors of stagnation which is related largely to absorption, distribution, metabolism and excretion (ADME) properties and various toxicities (T). ADME covers the pharmacokinetic issues determining whether a drug molecule gets to the target protein in the body, and how long it stays in the bloodstream. Parallel evaluation of efficiency and pharmacological properties of drug candidates has been standardized, and studies of ADMET processes are nowadays routinely carried out at early stage of drug discovery to reduce the attrition rate.

**Experimental setup.** We use 22 ADMET datasets in the TDC—the largest public benchmark for ADMET profiling to date. Endpoints in these datasets include metabolism with diverse types of CYP enzymes, half-life, clearance, and off-target effects. Real-world discovery studies drug candidates with diverse structures, and the ADMET benchmark datasets represent distribution shifts faced in

Table 1: **List of 66 datasets in Therapeutics Data Commons.** Size is the number of data points; Feature is the type of data features; Task is the type of prediction task; Metric is the recommended performance metric; Split is the recommended dataset split. For units, '—' is used to denote the that dataset defines either a classification task or a regression task for which numeric label units are not meaningful. For `generation.MolGen`, generic metrics are no applicable as performance is defined based on the task of interest.

| Dataset | Learning Task | Size | Unit | Feature | Task | Rec. Metric | Rec. Split |
|---|---|---|---|---|---|---|---|
| TDC.Caco2_Wang | single_pred.ADME | 906 | cm/s | Seq/Graph | Regression | MAE | Scaffold |
| TDC.HIA_Hou | single_pred.ADME | 578 | — | Seq/Graph | Binary | AUROC | Scaffold |
| TDC.Pgp_Broccatelli | single_pred.ADME | 1,212 | — | Seq/Graph | Binary | AUROC | Scaffold |
| TDC.Bioavailability_Ma | single_pred.ADME | 640 | — | Seq/Graph | Binary | AUROC | Scaffold |
| TDC.Lipophilicity_AstraZeneca | single_pred.ADME | 4,200 | log-ratio | Seq/Graph | Regression | MAE | Scaffold |
| TDC.Solubility_AqSolDB | single_pred.ADME | 9,982 | log-mol/L | Seq/Graph | Regression | MAE | Scaffold |
| TDC.BBB_Martins | single_pred.ADME | 1,975 | — | Seq/Graph | Binary | AUROC | Scaffold |
| TDC.PPBR_AZ | single_pred.ADME | 1,797 | % | Seq/Graph | Regression | MAE | Scaffold |
| TDC.VDss_Lombardo | single_pred.ADME | 1,130 | L/kg | Seq/Graph | Regression | Spearman | Scaffold |
| TDC.CYP2C19_Veith | single_pred.ADME | 12,092 | — | Seq/Graph | Binary | AUPRC | Scaffold |
| TDC.CYP2D6_Veith | single_pred.ADME | 13,130 | — | Seq/Graph | Binary | AUPRC | Scaffold |
| TDC.CYP3A4_Veith | single_pred.ADME | 12,328 | — | Seq/Graph | Binary | AUPRC | Scaffold |
| TDC.CYP1A2_Veith | single_pred.ADME | 12,579 | — | Seq/Graph | Binary | AUPRC | Scaffold |
| TDC.CYP2C9_Veith | single_pred.ADME | 12,092 | — | Seq/Graph | Binary | AUPRC | Scaffold |
| TDC.CYP2C9_Substrate | single_pred.ADME | 666 | — | Seq/Graph | Binary | AUPRC | Scaffold |
| TDC.CYP2D6_Substrate | single_pred.ADME | 664 | — | Seq/Graph | Binary | AUPRC | Scaffold |
| TDC.CYP3A4_Substrate | single_pred.ADME | 667 | — | Seq/Graph | Binary | AUROC | Scaffold |
| TDC.Half_Life_Obach | single_pred.ADME | 667 | hr | Seq/Graph | Regression | Spearman | Scaffold |
| TDC.Clearance_Hepatocyte_AZ | single_pred.ADME | 1,020 | $uL.min^{-1}.(10^6 cells)^{-1}$ | Seq/Graph | Regression | Spearman | Scaffold |
| TDC.Clearance_Microsome_AZ | single_pred.ADME | 1,102 | $mL.min^{-1}.g^{-1}$ | Seq/Graph | Regression | Spearman | Scaffold |
| TDC.LD50_Zhu | single_pred.Tox | 7,385 | log(1/(mol/kg)) | Seq/Graph | Regression | MAE | Scaffold |
| TDC.hERG | single_pred.Tox | 648 | — | Seq/Graph | Binary | AUROC | Scaffold |
| TDC.AMES | single_pred.Tox | 7,255 | — | Seq/Graph | Binary | AUROC | Scaffold |
| TDC.DILI | single_pred.Tox | 475 | — | Seq/Graph | Binary | AUROC | Scaffold |
| TDC.Skin_Reaction | single_pred.Tox | 404 | — | Seq/Graph | Binary | AUROC | Scaffold |
| TDC.Carcinogens_Lagunin | single_pred.Tox | 278 | — | Seq/Graph | Binary | AUROC | Scaffold |
| TDC.Tox21 | single_pred.Tox | 7,831 | — | Seq/Graph | Binary | AUROC | Scaffold |
| TDC.ClinTox | single_pred.Tox | 1,484 | — | Seq/Graph | Binary | AUROC | Scaffold |
| TDC.SARSCoV2_Vitro_Touret | single_pred.HTS | 1,480 | — | Seq/Graph | Binary | AUPRC | Scaffold |
| TDC.SARSCoV2_3CLPro_Diamond | single_pred.HTS | 879 | — | Seq/Graph | Binary | AUPRC | Scaffold |
| TDC.HIV | single_pred.HTS | 41,127 | — | Seq/Graph | Binary | AUPRC | Scaffold |
| TDC.QM7b | single_pred.QM | 7,211 | $eV/\beta^3$ | Coulomb | Regression | MAE | Random |
| TDC.QM8 | single_pred.QM | 21,786 | eV | Coulomb | Regression | MAE | Random |
| TDC.QM9 | single_pred.QM | 133,885 | $GHz/D/\beta_0^2/\beta_0^3$ | Coulomb | Regression | MAE | Random |
| TDC.USPTO_Yields | single_pred.Yields | 853,638 | % | Seq/Graph | Regression | MAE | Random |
| TDC.Buchwald-Hartwig | single_pred.Yields | 55,370 | % | Seq/Graph | Regression | MAE | Random |
| TDC.SAbDab_Liberis | single_pred.Paratope | 1,023 | — | Seq | Token-Binary | Avg-AUROC | Random |
| TDC.IEDB_Jespersen | single_pred.Epitope | 3,159 | — | Seq | Token-Binary | Avg-AUROC | Random |
| TDC.PDB_Jespersen | single_pred.Epitope | 447 | — | Seq | Token-Binary | Avg-AUROC | Random |
| TDC.TAP | single_pred.Develop | 242 | — | Seq | Regression | MAE | Random |
| TDC.SAbDab_Chen | single_pred.Develop | 2,409 | — | Seq | Regression | MAE | Random |
| TDC.Leenay | single_pred.CRISPROutcome | 1,521 | #/%/bits | Seq | Regression | MAE | Random |
| TDC.BindingDB_Kd | multi_pred.DTI | 52,284 | nM | Seq/Graph | Regression | MAE | Cold-start |
| TDC.BindingDB_IC50 | multi_pred.DTI | 991,486 | nM | Seq/Graph | Regression | MAE | Cold-start |
| TDC.BindingDB_Ki | multi_pred.DTI | 375,032 | nM | Seq/Graph | Regression | MAE | Cold-start |
| TDC.DAVIS | multi_pred.DTI | 27,621 | nM | Seq/Graph | Regression | MAE | Cold-start |
| TDC.KIBA | multi_pred.DTI | 118,036 | — | Seq/Graph | Regression | MAE | Cold-start |
| TDC.DrugBank_DDI | multi_pred.DDI | 191,808 | — | Seq/Graph | Multi-class | Macro-F1 | Random |
| TDC.TWOSIDES | multi_pred.DDI | 4,649,441 | — | Seq/Graph | Multi-label | Avg-AUROC | Random |
| TDC.HuRI | multi_pred.PPI | 51,813 | — | Seq | Binary | AUROC | Random |
| TDC.DisGeNET | multi_pred.GDA | 52,476 | — | Numeric/Text | Regression | MAE | Random |
| TDC.GDSC1 | multi_pred.DrugRes | 177,310 | $\mu M$ | Seq/Graph/Numeric | Regression | MAE | Random |
| TDC.GDSC2 | multi_pred.DrugRes | 92,703 | $\mu M$ | Seq/Graph/Numeric | Regression | MAE | Random |
| TDC.DrugComb | multi_pred.DrugSyn | 297,098 | — | Seq/Graph/Numeric | Regression | MAE | Combination |
| TDC.OncoPolyPharmacology | multi_pred.DrugSyn | 23,052 | — | Seq/Graph/Numeric | Regression | MAE | Combination |
| TDC.MHC1_IEDB-IMGT_Nielsen | multi_pred.PeptideMHC | 185,985 | log-ratio | Seq/Numeric | Regression | MAE | Random |
| TDC.MHC2_IEDB_Jensen | multi_pred.PeptideMHC | 134,281 | log-ratio | Seq/Numeric | Regression | MAE | Random |
| TDC.Protein_SAbDab | multi_pred.AntibodyAff | 493 | $K_D(M)$ | Seq/Numeric | Regression | MAE | Random |
| TDC.miRTarBase | multi_pred.MTI | 400,082 | — | Seq/Numeric | Regression | MAE | Random |
| TDC.USPTO_Catalyst | multi_pred.Catalyst | 721,799 | — | Seq/Graph | Multi-class | Macro-F1 | Random |
| TDC.MOSES | generation.MolGen | 1,936,962 | — | Seq/Graph | Generation | — | — |
| TDC.ZINC | generation.MolGen | 249,455 | — | Seq/Graph | Generation | — | — |
| TDC.ChEMBL | generation.MolGen | 1,961,462 | — | Seq/Graph | Generation | — | — |
| TDC.USPTO-50K | generation.RetroSyn | 50,036 | — | Seq/Graph | Generation | Top-K Acc | Random |
| TDC.USPTO_RetroSyn | generation.RetroSyn | 1,939,253 | — | Seq/Graph | Generation | Top-K Acc | Random |
| TDC.USPTO_Reaction | generation.Reaction | 1,939,253 | — | Seq/Graph | Generation | Top-K Acc | Random |

the wild. ADMET prediction requires models to generalize to domains unseen during training, i.e., molecules with a new scaffold structure that are structurally different from drugs used for training. To this end, we adopt scaffold split to simulate this distant effect. Each dataset is split into 7:1:2 training:validation:testing ratio where the training and validation sets are shuffled to create five random runs. For binary classification, the AUROC is used for balanced datasets and AUPRC for scenarios with fewer positive examples than negatives. For regression, we use the MAE. We use Spearman's rank correlation coefficient when rank-ordering of predictions is more important than the absolute error.

**Baselines.** The focus is on representation learning of molecular graphs. We include (1) multi-layer perceptron (MLP) with expert-curated fingerprints (Morgan fingerprint [120] with 1,024 bits) or descriptors (RDKit2D [79], 200-dim); (2) convolutional neural network (CNN) on SMILES strings, which applies 1D convolution over a string representation of the molecule [54]; (3) state-of-the-art (SOTA) models use graph neural networks on molecular 2D graphs, including neural fingerprint (NeuralFP) [37], graph convolutional network (GCN) [67], and attentive fingerprints (AttentiveFP) [163]. Further, [53] developed a pre-training strategy for molecular graphs, and we include two strategies, attribute masking (AttMasking) and context prediction (ContextPred), in our experiments. We select hyperparameters following recommendations in reference publications.

Table 2: **Results for the ADMET Benchmark Group.** Shown is average performance and standard deviation across five independent runs. Arrows (↑, ↓) indicate the direction of better performance. The best method is bolded and the second best is underlined.

| Raw Feature Type | | Expert-Curated Methods | | SMILES | Molecular Graph-Based Methods | | | | |
|---|---|---|---|---|---|---|---|---|---|
| Dataset | Metric | Morgan [120] | RDKit2D [79] | CNN [54] | NeuralFP [37] | GCN [67] | AttentiveFP [163] | AttrMasking [53] | ContextPred [53] |
| | # Params. | 1477K | 633K | 227K | 480K | 192K | 301K | 2067K | 2067K |
| TDC.Caco2 (↓) | MAE | $0.908_{\pm0.060}$ | $\mathbf{0.393_{\pm0.024}}$ | $0.446_{\pm0.036}$ | $0.530_{\pm0.102}$ | $0.599_{\pm0.104}$ | $\underline{0.401_{\pm0.032}}$ | $0.546_{\pm0.052}$ | $0.502_{\pm0.036}$ |
| TDC.HIA (↑) | AUROC | $0.807_{\pm0.072}$ | $0.972_{\pm0.008}$ | $0.869_{\pm0.026}$ | $0.943_{\pm0.014}$ | $0.936_{\pm0.024}$ | $\underline{0.974_{\pm0.007}}$ | $\mathbf{0.978_{\pm0.006}}$ | $0.975_{\pm0.004}$ |
| TDC.Pgp (↑) | AUROC | $0.880_{\pm0.006}$ | $0.918_{\pm0.007}$ | $0.908_{\pm0.012}$ | $0.902_{\pm0.020}$ | $0.895_{\pm0.021}$ | $0.892_{\pm0.012}$ | $\mathbf{0.929_{\pm0.006}}$ | $\underline{0.923_{\pm0.005}}$ |
| TDC.Bioav (↑) | AUROC | $0.581_{\pm0.086}$ | $\mathbf{0.672_{\pm0.021}}$ | $0.613_{\pm0.013}$ | $0.632_{\pm0.036}$ | $0.566_{\pm0.115}$ | $0.632_{\pm0.039}$ | $0.577_{\pm0.087}$ | $\underline{0.671_{\pm0.026}}$ |
| TDC.Lipo (↓) | MAE | $0.701_{\pm0.009}$ | $0.574_{\pm0.017}$ | $0.743_{\pm0.020}$ | $0.563_{\pm0.023}$ | $\underline{0.541_{\pm0.011}}$ | $0.572_{\pm0.007}$ | $0.547_{\pm0.024}$ | $\mathbf{0.535_{\pm0.012}}$ |
| TDC.AqSol (↓) | MAE | $1.203_{\pm0.019}$ | $\underline{0.827_{\pm0.047}}$ | $1.023_{\pm0.023}$ | $0.947_{\pm0.016}$ | $0.907_{\pm0.020}$ | $\mathbf{0.776_{\pm0.008}}$ | $1.026_{\pm0.020}$ | $1.040_{\pm0.045}$ |
| TDC.BBB (↑) | AUROC | $0.823_{\pm0.015}$ | $0.889_{\pm0.016}$ | $0.781_{\pm0.030}$ | $0.836_{\pm0.009}$ | $0.842_{\pm0.016}$ | $0.855_{\pm0.011}$ | $\underline{0.892_{\pm0.012}}$ | $\mathbf{0.897_{\pm0.004}}$ |
| TDC.PPBR (↓) | MAE | $12.848_{\pm0.362}$ | $9.994_{\pm0.319}$ | $11.106_{\pm0.358}$ | $\mathbf{9.292_{\pm0.384}}$ | $10.194_{\pm0.373}$ | $\underline{9.373_{\pm0.335}}$ | $10.075_{\pm0.202}$ | $9.445_{\pm0.224}$ |
| TDC.VD (↑) | Spearman | $0.493_{\pm0.011}$ | $\mathbf{0.561_{\pm0.025}}$ | $0.226_{\pm0.114}$ | $0.258_{\pm0.162}$ | $0.457_{\pm0.050}$ | $0.241_{\pm0.145}$ | $\underline{0.559_{\pm0.019}}$ | $0.485_{\pm0.092}$ |
| TDC.CYP2D6-I (↑) | AUPRC | $0.587_{\pm0.011}$ | $0.616_{\pm0.007}$ | $0.544_{\pm0.053}$ | $0.627_{\pm0.009}$ | $0.616_{\pm0.020}$ | $0.646_{\pm0.014}$ | $\underline{0.721_{\pm0.009}}$ | $\mathbf{0.739_{\pm0.005}}$ |
| TDC.CYP3A4-I (↑) | AUPRC | $0.827_{\pm0.009}$ | $0.829_{\pm0.007}$ | $0.821_{\pm0.003}$ | $0.849_{\pm0.004}$ | $0.840_{\pm0.010}$ | $0.851_{\pm0.006}$ | $\underline{0.902_{\pm0.002}}$ | $\mathbf{0.904_{\pm0.002}}$ |
| TDC.CYP2C9-I (↑) | AUPRC | $0.715_{\pm0.004}$ | $0.742_{\pm0.006}$ | $0.713_{\pm0.006}$ | $0.739_{\pm0.010}$ | $0.735_{\pm0.004}$ | $0.749_{\pm0.004}$ | $\underline{0.829_{\pm0.003}}$ | $\mathbf{0.839_{\pm0.003}}$ |
| TDC.CYP2D6-S (↑) | AUROC | $0.671_{\pm0.066}$ | $0.677_{\pm0.047}$ | $0.485_{\pm0.037}$ | $0.572_{\pm0.062}$ | $0.617_{\pm0.039}$ | $0.574_{\pm0.030}$ | $\underline{0.704_{\pm0.028}}$ | $\mathbf{0.736_{\pm0.024}}$ |
| TDC.CYP3A4-S (↑) | AUROC | $0.633_{\pm0.013}$ | $\underline{0.639_{\pm0.012}}$ | $\mathbf{0.662_{\pm0.031}}$ | $0.578_{\pm0.020}$ | $0.590_{\pm0.023}$ | $0.576_{\pm0.025}$ | $0.582_{\pm0.021}$ | $0.609_{\pm0.025}$ |
| TDC.CYP2C9-S (↑) | AUPRC | $0.380_{\pm0.015}$ | $0.360_{\pm0.040}$ | $0.367_{\pm0.059}$ | $0.359_{\pm0.059}$ | $0.344_{\pm0.051}$ | $0.375_{\pm0.032}$ | $\underline{0.381_{\pm0.045}}$ | $\mathbf{0.392_{\pm0.026}}$ |
| TDC.Half_Life (↑) | Spearman | $\mathbf{0.329_{\pm0.083}}$ | $0.184_{\pm0.111}$ | $0.038_{\pm0.138}$ | $0.177_{\pm0.165}$ | $0.239_{\pm0.100}$ | $0.085_{\pm0.068}$ | $0.151_{\pm0.068}$ | $0.129_{\pm0.114}$ |
| TDC.CL-Micro (↑) | Spearman | $0.492_{\pm0.020}$ | $\mathbf{0.586_{\pm0.014}}$ | $0.252_{\pm0.116}$ | $0.529_{\pm0.015}$ | $0.532_{\pm0.033}$ | $0.365_{\pm0.055}$ | $\underline{0.585_{\pm0.034}}$ | $0.578_{\pm0.007}$ |
| TDC.CL-Hepa (↑) | Spearman | $0.272_{\pm0.068}$ | $0.382_{\pm0.007}$ | $0.235_{\pm0.021}$ | $0.401_{\pm0.037}$ | $0.366_{\pm0.063}$ | $0.289_{\pm0.022}$ | $\underline{0.413_{\pm0.028}}$ | $\mathbf{0.439_{\pm0.026}}$ |
| TDC.hERG (↑) | AUROC | $0.736_{\pm0.023}$ | $\mathbf{0.841_{\pm0.020}}$ | $0.754_{\pm0.037}$ | $0.722_{\pm0.034}$ | $0.738_{\pm0.038}$ | $\underline{0.825_{\pm0.007}}$ | $0.778_{\pm0.046}$ | $0.756_{\pm0.023}$ |
| TDC.AMES (↑) | AUROC | $0.794_{\pm0.008}$ | $0.823_{\pm0.011}$ | $0.776_{\pm0.015}$ | $0.823_{\pm0.006}$ | $0.818_{\pm0.010}$ | $0.814_{\pm0.008}$ | $\mathbf{0.842_{\pm0.008}}$ | $\underline{0.837_{\pm0.009}}$ |
| TDC.DILI (↑) | AUROC | $0.832_{\pm0.021}$ | $0.875_{\pm0.019}$ | $0.792_{\pm0.016}$ | $0.851_{\pm0.026}$ | $0.859_{\pm0.033}$ | $\underline{0.886_{\pm0.015}}$ | $\mathbf{0.919_{\pm0.008}}$ | $0.861_{\pm0.018}$ |
| TDC.LD50 (↓) | MAE | $0.649_{\pm0.019}$ | $\underline{0.678_{\pm0.003}}$ | $0.675_{\pm0.011}$ | $0.667_{\pm0.020}$ | $0.649_{\pm0.026}$ | $0.678_{\pm0.012}$ | $\mathbf{0.685_{\pm0.025}}$ | $0.669_{\pm0.030}$ |

**Results.** Results are shown in Table 2. Overall, we find that pre-training GIN (Graph Isomorphism Network) [165] with context prediction has the best performances across 8 endpoints, attribute masking performs best in 5 endpoints, with 13 combined for pre-training strategies and outstanding performance in the CYP enzyme predictions. Expert-curated molecular descriptors (RDKit2D) achieve the best results in five endpoints, while the SMILES-based CNN yields a best-performing predictor for one endpoint. Our benchmarking led to three key findings. First, the ML SOTA models do not work well consistently for these novel realistic endpoints. In some cases, methods based on learned features are worse than the efficient domain features. This gap highlights the necessity for realistic benchmarking. Second, performances vary across feature types given different endpoints. For example, on **TDC.CYP3A4-S** dataset, SMILES-based CNN model outperforms graph-based methods by 8.7%-14.9%. This result can be explained by heterogeneous information captured by different molecular representations; GNN models focus on local substructures of molecular graphs, whereas descriptors attend to global biochemical features. Thus, future integration of these diverse signals can further improve model performance. Third, best-performing methods use pre-training, highlighting a potentially fruitful future direction for self-supervised learning.

## 4.2 The Challenge of Domain Generalization in the Drug-Target Interaction Benchmark

**Motivation.** Drug-target interactions (DTI) characterize the binding affinity of compounds to target molecules. Despite promising prediction accuracies of supervised computational models for DTI prediction [54], their use in practice, such as for novel drugs in development, is hindered by the assumption that there are already known and similar drugs for a given target of interest. In particular, those models adopt random dataset splits—while the testing set contains compound-target pairs unseen during training, both the compound and the target molecule are represented in the training set, albeit in different molecular combinations. This pitfall of existing evaluation strategies becomes apparent when the models are used in, for example, compound screening campaigns searching for novel target candidates or a novel class of compounds for known targets. Further, the models need to have the ability to generalize to new targets and compounds as their structural and biochemical characteristics shift over years of development, meaning that the models need to be robust to subtle domain shifts over time in order to be practically useful.

**Experimental setup.** We use DTIs in **TDC.BindingDB** and collate them with patent information on target discovery. In particular, we define data domains such data each domain consists of DTIs patented in a specific year. We evaluate domain generalization models to predict out-of-distribution DTIs between 2019-2021 after training the models on DTI data from 2013-2018, simulating real-world discovery. Because time information for specific targets and compounds can often be confidential, we use the patent year of the DTI as a reasonable proxy. We consider a popular DeepDTA model [107] as the backbone of domain generalization algorithms. The evaluation metric is Pearson's correlation coefficient (PCC). Selection of the validation set is crucial for a fair comparison of domain generalization methods. We follow the strategy of "Training-domain validation set" from [46] and proceed as follows. Using DTI information from 2013-2018, we randomly select 20% DTIs as a validation

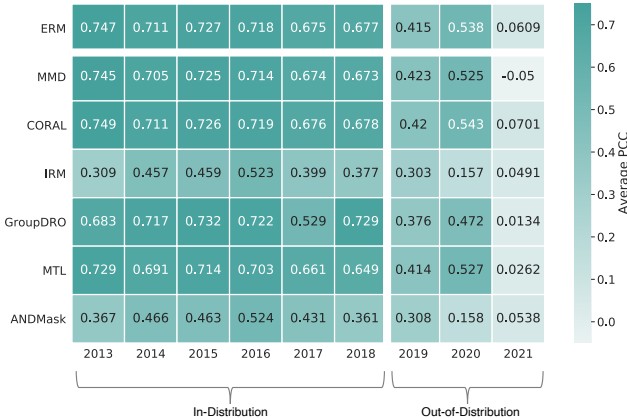

Figure 4: **Heatmap visualization of domain generalization performance across domains in the DTI-DG benchmark using** TDC.BindingDB. We observe a significant gap between in-distribution and out-of-distribution performance, indicating the limited ability of existing models to extrapolate to more complicated patterns.

Table 3: **Results on the DTI-DG benchmark using** TDC.BindingDB. "In-Dist." combines the in-split validation set and has similar data distribution as the training set (2013-2018). "Out-Dist." aggregates testing domains (2019-2021). The goal is to maximize performance on the testing domains. Shown are average and standard deviation values of Pearson's Correlation Coefficient across five random runs. The best method is bolded and the second best is underlined.

| Method | In-Dist. | Out-Dist. |
|---|---|---|
| ERM | 0.703±0.005 | 0.427±0.012 |
| MMD | 0.700±0.002 | **0.433±0.010** |
| CORAL | **0.704±0.003** | 0.432±0.010 |
| IRM | 0.420±0.008 | 0.284±0.021 |
| GroupDRO | 0.681±0.010 | 0.384±0.006 |
| MTL | 0.685±0.009 | 0.425±0.010 |
| ANDMask | 0.436±0.014 | 0.288±0.019 |

set and use them for in-distribution performance calculation because they represent a distribution over data similar to the training set. We use DTI information from 2018-2021 only during testing and refer to it as "out-of-distribution performance."

**Baselines.** ERM (Empirical Risk Minimization) [150] is a standard training strategy simultaneously minimizing errors across all data domains. We include the following domain generalization algorithms: MMD (Maximum Mean Discrepancy) [83] optimizes the similarities of maximum mean discrepancy across domains, CORAL (Correlation Alignment) [137] matches the mean and the covariance of features across domains; IRM (Invariant Risk Minimization) [5] generates features using a linear classifier across domains; GroupDRO (distributionally robust neural networks for group shifts) [123] optimizes ERM and adjusts weights of domains with larger errors; MTL (Marginal Transfer Learning) [19] concatenates the original features with augmented vectors representing marginal distributions of feature vectors; ANDMask [108] masks gradients with inconsistent signs in corresponding weights across domains. Most of these methods are developed for classification; we adapt them for regression and keep the rest of the model architecture the same. We use the default hyperparameters described in reference publications.

**Results.** Results are shown in Table 3 and Figure 4. We find that in-distribution performance reaches 0.7 PCC and is stable across years, suggesting robust predictive power of existing models in widely adopted yet unrealistic evaluation scenarios. However, the out-of-distribution performance significantly degrades from 33.9% to 43.6% across methods, suggesting that domain shifts break prevailing training strategies. Second, while the best-performed methods are MMD and CORAL, standard training strategy achieves similar performances as state-of-the-art domain generalization methods, which is in agreement with a systematic study conducted by [46], highlighting the need for robust domain generalization methods.

### 4.3 The Challenge of Molecule Generation in the DRD3 Docking Benchmark

**Motivation.** AI-assisted drug design aims to generate molecular structures with desired biological properties. Despite recent advances in generative modeling, existing methods in this area optimize ad-hoc heuristic oracles, such as QED (quantitative estimate of drug-likeness) and LogP (Octanol-water partition coefficient) [63, 169, 174]. Further, laboratory experiments, such as bioassays and high-fidelity simulations like molecular docking, are resource-intensive, thus creating a need for data-efficient generative models. The low-resource constraints suggest that the number of oracle calls available to a generative model should be limited; however, this aspect is ignored by existing models, which typically rely on millions of oracle calls to generate a molecule with desired biological properties [174, 169].

Motivated by this open and difficult challenge, we consider molecular docking [28, 134] as an example of a high-quality oracle that is also resource-intensive. In particular, it takes only a few milliseconds for an oracle such as QED to provide an answer to the generative model; however, it can

take up to 5 seconds for docking (using vina on a CPU). Docking evaluates the affinity between a ligand (such as a small molecule drug) and a candidate target (such as a protein or enzyme) and is widely used in real-world drug discovery [93] and considerably more informative than simple oracles like the QED. Further, generative models can generate molecules with structure outside of pre-defined chemical space, meaning that the generated molecule might have a valid chemical structure but could not be practically synthesized in a laboratory [40]. For this reason, we here consider pre-defined domain filters and the synthetic accessibility score to evaluate the quality of generated molecules in addition to the above-mentioned frequency of the oracle access. We proceed with the description of the generation benchmark in TDC.

**Experimental setup.** We use **TDC.ZINC** dataset as the molecule library and **TDC.Docking** oracle function as the docking score evaluator against the target protein DRD3, which is a target for neurology diseases such as tremor and schizophrenia. To imitate low-data scenarios, we limit the number of oracle calls available to each model to either 100, 500, 1000, or 5000 calls. In addition to oracle scores, we investigate the following metrics to evaluate the quality of generated molecules: (1) *Top100/Top10/Top1* is the average docking score of top-100/10/1 molecules generated for a given target; (2) *Diversity* is the average pairwise Tanimoto distance of Morgan fingerprints for top-100 generated molecules; (3) *Novelty* is the fraction of generated molecules that are not present in the training set; (4) *m1* is the synthesizability score of molecules obtained via molecule.one retrosynthesis model [122]; (5) *%pass* is the fraction of generated molecules that successfully pass through a set of pre-defined filters; (6)*Top1 %pass* is the lowest docking score for molecules that pass the filter. Every model is run three times with different random seeds.

**Baselines.** We consider the following domain SOTA methods: Screening (simulated as random sampling) [93], Graph-GA (graph-based genetic algorithm) [58], and the following ML SOTA methods: string-based LSTM [129], GCPN (Graph Convolutional Policy Network) [169], MolDQN (Deep Q-Network) [174], and MARS (Markov molecular Sampling) [162]. As a reference, we include *best-in-data* strategy, which chooses 100 molecules with the highest docking score from the ZINC 250K database. We select hyperparameters as described in reference publications. We train models with different random seeds and report average performance and standard deviation across three independent runs.

**Results.** Results are shown in Table 4. Overall, we find that no existing model performs well in challenging oracle scenarios. In particular, most methods do not surpass the best-in-data docking scores in scenarios with 100, 500, or 1,000 allowable oracle calls except for Graph-GA (-14.811) and LSTM (-13.017) models that outperform the best-in-data reference but can do so only in the scenario with 5,000 oracle calls. Considering optimization ability, Graph-GA dominates the leaderboard with zero trainable parameters, while a simple SMILES LSTM model ranks behind. Thus, while SOTA ML models achieve strong performances in unlimited oracle scenarios, they cannot beat virtual screening when they are allowed to perform at most 5,000 oracle calls. This finding raises concerns regarding the utility of SOTA ML methods and calls for a shift of focus in molecular generation research to consider real-world constraints in model evaluation.

As for synthesizability, as the number of allowable oracle calls grows, the more significant fraction of generated molecules have undesired structures despite increasing affinity scores. We observe a monotonous increase in the m1 score for the best-performing Graph GA model when allowing more oracle calls. In the scenario with 5,000 oracle calls, only 2.3% - 52.7% of generated molecules successfully pass through quality filters. The best docking score significantly drops when considering only the set of molecules that pass through the filters. In contrast, the LSTM model generates molecules with relatively good quality across all categories, indicating that generative models can better capture the distribution of molecules in a training set to produce molecules that can likely be synthesized in a laboratory. To address this problem, synthesizable constrained generation approaches [71, 44, 21] represent a promising future strategy.

## 5    Conclusion

The attention of the machine learning community to therapeutics remains relatively limited, compared to areas such as natural language processing and computer vision, even though therapeutics offer many challenging algorithmic problems and applications of immense impact. To this end, our Therapeutics Data Commons (TDC) is a platform of AI-ready datasets and learning tasks for drug

Table 4: **Results on the DRD3 docking benchmark using TDC.ZINC and TDC.Docking datasets.** Shown are average and standard deviation values across three independent runs. Arrows (↑, ↓) indicate the direction of better performance. The best method is bolded and the second best is underlined.

| Method Category | | | Domain-Specific Methods | | State-of-the-art ML Methods | | | |
|---|---|---|---|---|---|---|---|---|
| Metric | Best-in-data | # Calls | Screening [93] | Graph-GA [58] | LSTM [129] | GCPN [169] | MolDQN [174] | MARS [162] |
| # Params. | - | - | 0 | 0 | 3149K | 18K | 2694K | 153K |
| Top100 (↓) | -12.080 | | $-7.554_{\pm0.065}$ | $-7.222_{\pm0.013}$ | $\mathbf{-7.594_{\pm0.182}}$ | $3.860_{\pm0.102}$ | $-5.178_{\pm0.341}$ | $-5.928_{\pm0.298}$ |
| Top10 (↓) | -12.590 | | $-9.727_{\pm0.276}$ | $\mathbf{-10.177_{\pm0.158}}$ | $-10.033_{\pm0.186}$ | $-5.617_{\pm0.413}$ | $-6.438_{\pm0.176}$ | $-8.133_{\pm0.328}$ |
| Top1 (↓) | -12.800 | | $-10.367_{\pm0.464}$ | $\mathbf{-11.767_{\pm1.087}}$ | $-11.133_{\pm0.634}$ | $-11.633_{\pm2.217}$ | $-7.020_{\pm0.194}$ | $-9.100_{\pm0.712}$ |
| Diversity (↑) | 0.864 | 100 | $0.881_{\pm0.002}$ | $0.885_{\pm0.001}$ | $0.884_{\pm0.002}$ | $\mathbf{0.909_{\pm0.001}}$ | $0.907_{\pm0.001}$ | $0.873_{\pm0.010}$ |
| Novelty (↑) | - | | - | $1.000_{\pm0.000}$ | $1.000_{\pm0.000}$ | $1.000_{\pm0.000}$ | $1.000_{\pm0.000}$ | $1.000_{\pm0.000}$ |
| %Pass (↑) | 0.780 | | $0.717_{\pm0.005}$ | $0.693_{\pm0.037}$ | $0.763_{\pm0.019}$ | $0.093_{\pm0.009}$ | $0.017_{\pm0.012}$ | $\mathbf{0.807_{\pm0.033}}$ |
| Top1 Pass (↓) | -11.700 | | $-2.467_{\pm2.229}$ | $0.000_{\pm0.000}$ | $-1.100_{\pm1.417}$ | $7.667_{\pm0.262}$ | $-3.630_{\pm2.588}$ | $\mathbf{-3.633_{\pm0.946}}$ |
| m1 (↓) | 5.100 | | $4.845_{\pm0.235}$ | $5.223_{\pm0.256}$ | $5.219_{\pm0.247}$ | $10.000_{\pm0.000}$ | $10.000_{\pm0.000}$ | $\mathbf{4.470_{\pm1.047}}$ |
| Top100 (↓) | -12.080 | | $-9.341_{\pm0.039}$ | $\mathbf{-10.036_{\pm0.221}}$ | $-9.419_{\pm0.173}$ | $-8.119_{\pm0.104}$ | $-6.357_{\pm0.084}$ | $-7.278_{\pm0.198}$ |
| Top10 (↓) | -12.590 | | $-10.517_{\pm0.135}$ | $\mathbf{-11.527_{\pm0.533}}$ | $-10.687_{\pm0.335}$ | $-10.230_{\pm0.354}$ | $-7.173_{\pm0.166}$ | $-9.067_{\pm0.377}$ |
| Top1 (↓) | -12.800 | | $-11.167_{\pm0.309}$ | $\mathbf{-12.500_{\pm0.748}}$ | $-11.367_{\pm0.579}$ | $-11.967_{\pm0.680}$ | $-7.620_{\pm0.185}$ | $-9.833_{\pm0.309}$ |
| Diversity (↑) | 0.864 | 500 | $0.870_{\pm0.003}$ | $0.857_{\pm0.005}$ | $0.875_{\pm0.005}$ | $\mathbf{0.914_{\pm0.001}}$ | $0.903_{\pm0.002}$ | $0.866_{\pm0.005}$ |
| Novelty (↑) | - | | - | $1.000_{\pm0.000}$ | $1.000_{\pm0.000}$ | $1.000_{\pm0.000}$ | $1.000_{\pm0.000}$ | $1.000_{\pm0.000}$ |
| %Pass (↑) | 0.780 | | $\mathbf{0.770_{\pm0.029}}$ | $0.710_{\pm0.080}$ | $0.727_{\pm0.012}$ | $0.127_{\pm0.005}$ | $0.030_{\pm0.016}$ | $0.660_{\pm0.050}$ |
| Top1 Pass (↓) | -11.700 | | $-8.767_{\pm0.047}$ | $\mathbf{-9.300_{\pm0.163}}$ | $-8.767_{\pm0.170}$ | $-7.200_{\pm0.141}$ | $-6.030_{\pm0.073}$ | $-6.100_{\pm0.141}$ |
| m1 (↓) | 5.100 | | $\mathbf{5.672_{\pm1.211}}$ | $6.493_{\pm0.341}$ | $5.787_{\pm0.934}$ | $10.000_{\pm0.000}$ | $10.000_{\pm0.000}$ | $5.827_{\pm0.937}$ |
| Top100 (↓) | -12.080 | | $-9.693_{\pm0.019}$ | $\mathbf{-11.224_{\pm0.484}}$ | $-9.971_{\pm0.115}$ | $-9.053_{\pm0.080}$ | $-6.738_{\pm0.042}$ | $-8.224_{\pm0.196}$ |
| Top10 (↓) | -12.590 | | $-10.777_{\pm0.189}$ | $\mathbf{-12.400_{\pm0.782}}$ | $-11.163_{\pm0.141}$ | $-11.027_{\pm0.273}$ | $-7.506_{\pm0.085}$ | $-9.843_{\pm0.068}$ |
| Top1 (↓) | -12.800 | | $-11.500_{\pm0.432}$ | $\mathbf{-13.233_{\pm0.713}}$ | $-11.967_{\pm0.205}$ | $-12.033_{\pm0.618}$ | $-7.800_{\pm0.042}$ | $-11.100_{\pm0.141}$ |
| Diversity (↑) | 0.864 | 1000 | $0.873_{\pm0.003}$ | $0.815_{\pm0.046}$ | $0.871_{\pm0.004}$ | $\mathbf{0.913_{\pm0.001}}$ | $0.904_{\pm0.001}$ | $0.871_{\pm0.004}$ |
| Novelty (↑) | - | | - | $1.000_{\pm0.000}$ | $1.000_{\pm0.000}$ | $1.000_{\pm0.000}$ | $1.000_{\pm0.000}$ | $1.000_{\pm0.000}$ |
| %Pass (↑) | 0.780 | | $0.757_{\pm0.026}$ | $\mathbf{0.777_{\pm0.096}}$ | $0.777_{\pm0.026}$ | $0.170_{\pm0.022}$ | $0.033_{\pm0.005}$ | $0.563_{\pm0.052}$ |
| Top1 Pass (↓) | -11.700 | | $-9.167_{\pm0.047}$ | $\mathbf{-10.600_{\pm0.374}}$ | $-9.367_{\pm0.094}$ | $-8.167_{\pm0.047}$ | $-6.450_{\pm0.085}$ | $-7.367_{\pm0.205}$ |
| m1 (↓) | 5.100 | | $5.527_{\pm0.780}$ | $7.695_{\pm0.909}$ | $\mathbf{4.818_{\pm0.541}}$ | $10.000_{\pm0.000}$ | $10.000_{\pm0.000}$ | $6.037_{\pm0.137}$ |
| Top100 (↓) | -12.080 | | $-10.542_{\pm0.035}$ | $\mathbf{-14.811_{\pm0.413}}$ | $-13.017_{\pm0.385}$ | $-10.045_{\pm0.226}$ | $-8.236_{\pm0.089}$ | $-9.509_{\pm0.035}$ |
| Top10 (↓) | -12.590 | | $-11.483_{\pm0.056}$ | $\mathbf{-15.930_{\pm0.336}}$ | $-14.030_{\pm0.421}$ | $-11.483_{\pm0.581}$ | $-9.348_{\pm0.188}$ | $-10.693_{\pm0.172}$ |
| Top1 (↓) | -12.800 | | $-12.100_{\pm0.356}$ | $\mathbf{-16.533_{\pm0.309}}$ | $-14.533_{\pm0.525}$ | $-12.300_{\pm0.993}$ | $-9.990_{\pm0.194}$ | $-11.433_{\pm0.450}$ |
| Diversity (↑) | 0.864 | 5000 | $0.872_{\pm0.003}$ | $0.626_{\pm0.092}$ | $0.740_{\pm0.056}$ | $\mathbf{0.922_{\pm0.002}}$ | $0.893_{\pm0.005}$ | $0.873_{\pm0.002}$ |
| Novelty (↑) | - | | - | $1.000_{\pm0.000}$ | $1.000_{\pm0.000}$ | $1.000_{\pm0.000}$ | $1.000_{\pm0.000}$ | $1.000_{\pm0.000}$ |
| %Pass (↑) | 0.780 | | $\mathbf{0.683_{\pm0.073}}$ | $0.393_{\pm0.308}$ | $0.257_{\pm0.103}$ | $0.167_{\pm0.045}$ | $0.023_{\pm0.012}$ | $0.527_{\pm0.087}$ |
| Top1 Pass (↓) | -11.700 | | $-10.100_{\pm0.000}$ | $\mathbf{-14.267_{\pm0.450}}$ | $-12.533_{\pm0.403}$ | $-9.367_{\pm0.170}$ | $-7.980_{\pm0.112}$ | $-9.000_{\pm0.082}$ |
| m1 (↓) | 5.100 | | $\mathbf{5.610_{\pm0.805}}$ | $9.669_{\pm0.468}$ | $5.826_{\pm1.908}$ | $10.000_{\pm0.000}$ | $10.000_{\pm0.000}$ | $7.073_{\pm0.798}$ |

discovery and development. Curated datasets, strategies for systematic model development and evaluation, and an ecosystem of tools, leaderboards, and community resources in TDC serve as a meeting point for domain and machine learning scientists. We envision that TDC can considerably accelerate machine-learning model development, validation, and transition into implementation.

To facilitate algorithmic and scientific innovation in therapeutics, we will support the continued development of TDC to provide a software ecosystem with AI-ready datasets and enhance outreach to build an inclusive research community.

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
