# OpenReview forum: "Therapeutics Data Commons: Machine Learning Datasets and Tasks for Drug Discovery and Development"
_NeurIPS.cc/2021/Track/Datasets_and_Benchmarks/Round1 — NeurIPS 2021 Datasets and Benchmarks Track (Round 1)_

### Official Review · Reviewer_fHUr · 2021-07-02
**A drug discovery and development dataset hub with some Python tools.**

**Rating:** 6
**Confidence:** 4

**Strengths:**

* The authors have collected a good number drug discovery and development datasets that have been previously published.
* The authors created website and GitHub repositories for user to get access to the data and code.

**Weaknesses:**

* No novel dataset contribution to the community. All the datasets were collected from the previous publications or resources. No novel problems and tasks proposed.

* The authors were emphasizing what's different of their datasets from the other biochemical and chemical repositories are their datasets are data science ready. But it seems that many datasets were just tabular CSV datasets which are easy to preprocess, and the author did not mention how they make those datasets data science ready and whether it is easy or hard.

* TDC has to be maintained manually. There has to be people collecting new datasets from new publications and resources and arranging the new datasets. There is no easy way to consume new datasets automatically. I also did not find an interface to contribute new datasets on the TDC website, and all the dataset contributions have to go through the Slack channel. There is a lack of dataset contribution standards as well.

* I am not sure if the leaderboard can be truly trusted and useful. Instead of hiding the test set and hosting a server to evaluate the submissions from the users, TDC releases test sets and accepts user reported results via forms, although only the ones whose implementation is public and reproducible will be considered. This prevents the users, who do not want to release their code, from submitting their good results. I have to admit because of the nature of TDC that all the datasets were collected somewhere else and it is impossible to hide the test set. However, it does not make sense to serve a leaderboard unless TDC has its own private datasets and evaluation server for the leaderboard.

* Despite the TDC Python library has provided a couple of examples to get the user started, it does not have Sphinx API documentation or something equivalent, which prevents the user from using, extending, and contributing to the code. The code quality remains to be improved. The TDC Python library also implicitly assumes that the dataset size is small and can be handled by Pandas. In the age of data explosion, this assumption might not hold anymore. Other dataframe libraries, such as Rapids, might be more suitable for large datasets. Most of the data preprocessing and model evaluation metric implementation seems just trivial.

* TDC ecosystem is a concern. Most of the MR contributions so far are from the major authors of the paper. Without the contribution momentum, TDC’s future will be a concern.


**Additional Feedback:**

* TDC is a good collection of datasets used in the publications. However, it might not serve well for the people who wants to create custom datasets.
* The TDC Python library is a good start for rookies, but it might not be too useful for experts.
* I was expecting novel datasets and/or well-investigated benchmarking on the existing datasets. But I did not see them in the paper.

**Clarity:**

* Some of the abbreviations were not explained. For example, what is "GIN" (line 153)?
* Some pretraining (line 153) details were not given either.
* Many tables in the papers were not referenced in the text.
* How was the experiment repeated? What are the randomnesses of the runs?
* In chapter 4.2, the authors used temporal split, instead of other split methods, to split the drug-target interactions (DTI) dataset, without formulating the motivations clearly. The author should specifically mention that because there will be new chemical compounds being developed as drug and new proteins being investigated as targets, the model trained on the old dataset will not be valuable if it cannot perform well on unseen chemical compounds and proteins.


**Correctness:**

* I sort of disagree with some of the conclusions drawn from the benchmarks.
* According to the authors, the default hyperparameters were used for the models and methods they obtained from the community, and the author did not try too much to optimize the model specifically for the datasets being used. For example, if the dataset is very small, but the number of parameters in the model is very large, without rigorous regularization or early stopping methods, the model will likely overfit to the training data and perform poor on the test data.
* In chapter 4.1, the authors claimed that "the ML SOTA models do not work well consistently for these novel realistic endpoints." However, close examination on the result Table 2 reveals that for the benchmarks that SOTA ML methods did not perform the best, the difference between the top curated methods and the top SOTA ML methods are not too large. Not to mention the hyperparameters and model architectures of the ML models were not well explored.
* In chapter 4.3,  the authors were evaluating the methods on the molecule generation tasks under the constraint of limited oracle calls. In practice, however, depending on the oracle algorithm, parallel computing implementation, and the distributed computing resources, the number of oracle calls could be almost "unlimited". Large pharmaceutical companies also have huge computing resources for oracle calls. If unlimited oracle calls is truly infeasible in practice, the author would have to mention in the paper why that is the case.



**Documentation:**

* I think the authors have created sufficient details on data collection and organization, availability and maintenance, and ethical and responsible use. The authors claimed that the core development team is committed and has resources to maintain and actively develop TDC for at minimum the next five years, although I am not sure if five years is sufficiently long for the community.
* The authors provides Jupyter Notebook examples to get the user started.
* The authors provided the benchmark source code in the GitHub repository for the users to reproduce.
* The API documentation, however, is missing. The authors should annotate the code well and host the API documentation online.

**Ethics:**

* No very big deal, since there are no novel datasets. The only concern is whether the original dataset creators will allow TDC to redistribute the dataset.

**Relation To Prior Work:**

* Overall, it is clearly discussed how this work differs from previous contributions.
* One thing I have to mention is that I think the author should have mentioned [DrugBank](https://go.drugbank.com/), which is highly related to TDC, in the related work chapter. The DrugBank provides APIs for the user to curate existing drug information. Most of the TDC dataset information might have been covered by DrugBank.


**Summary And Contributions:**

* TDC provides a data science ready dataset hub that allows the users to explore and retrieve drug discovery and development datasets.
* A TDC Python library was created for data preprocessing and method evaluation to help the users get started quickly.
* Some open source ML methods were applied on the datasets to solve drug discovery and development problems, and their performance were compared with the old conventional methods.

* Rating modified on July 19th, 2021 from 4 to 6.

---

> ### Author Response · Authors · 2021-07-09
> **Re: Official Blind Reviewer fHUr - Part 1**
>
> We thank the reviewer for taking the time and effort to review our work and provide detailed feedback. We carefully checked the comments and provided our detailed response below.
>
> RE: No novel datasets.
>
> Many incredibly influential benchmarks in the ML community identify existing raw data resources (i.e., they do not generate “novel” resources or experimentally measure “novel” phenomena), process those resources, and use the processed datasets to formulate ML tasks to advance ML algorithms. Examples of such benchmarks include OGB (Hu et al., 2020, NeurIPS) hosts a set of graphs for which raw data previously existed in different domains, yet OGB has emerged as a leading benchmark for graph machine learning; WILDS (Koh et al., 2021, ICML) collects a diverse set of datasets of previously existing labels, yet it is a crucial benchmark for the problem of domain generalization; MLPerf (Mattson et al., 2020, MLSys) uses existing datasets such as ImageNet, COCO but it has become the standard benchmark for ML system comparison.
>
> Further, benchmarks defined on top of raw data are especially common in the realm of scientific discovery. This is because generating a single “novel” dataset with labeled examples can cost millions of dollars (e.g., confer Jeff Janes et al., 2018). For example, in the biomedical literature, TAPE (Rao et al., 2019, NeurIPS) curates five protein-related tasks from various existing protein databases, yet the TAPE benchmark has advanced protein representation learning algorithms; MoleculeNet (Wu et al., 2018, Chemical Science) collects 16 endpoints of molecular property predictions from various sources and it lays the foundation for recent advances in molecular machine learning.
>
> Similar to the above-mentioned benchmarks, TDC provides a collection of curated datasets and learning tasks in the area of therapeutics discovery and development. Most TDC datasets are completely new for the ML community and have previously not been used in machine learning research. By providing ML-ready datasets (i.e., dataset, task definition, dataset split, performance evaluation strategy), TDC can facilitate algorithmic advances in the broad area of therapeutics.
>
> RE: Concern about the future of the TDC ecosystem.
>
> That is an important question, thank you. While TDC is still a young initiative, it has been growing rapidly and is being actively maintained and developed by a team of volunteers. Since the initial release of TDC 6 months ago, we have seen:
>
> * over 17K downloads of TDC datasets (https://doi.org/10.7910/DVN/21LKWG)
> * over 16K downloads of TDC package (https://pepy.tech/project/pytdc)
> * over 2.8K downloads of TDC package per month (https://pepy.tech/project/pytdc).
>
> These usage statistics suggest that TDC has a high potential to continue to grow in the future.
>
> Further, to continually support the growth of the TDC ecosystem, we have recently recruited a community manager (https://tdcommons.ai/team). We are also planning the first TDC user group meeting at the end of the summer to facilitate contributions from the growing community of TDC users. Finally, we note that we use the merge request (MR) feature in Github to keep track of features added within the team and each MR needs to be approved by a core TDC member in order for the MR to be merged into the master branch. Because of that, MR might not be a good measure of the ecosystem's growth. We hope the quick growth of the TDC user community and the above actions provide sufficient evidence that the future of TDC is not a concern.

---

> > ### Author Response · Authors · 2021-07-09
> > **Re: Official Blind Reviewer fHUr - Part 2**
> >
> > RE: Many datasets were just tabular CSV datasets that are easy to preprocess.
> >
> > The reviewer raised a concern that “some datasets are just tabular CSV datasets that are easy to preprocess, and the authors do not mention how they make those datasets data science ready and whether that is easy or hard.” While we provide further details on dataset construction in the Supplementary Document, we acknowledge that we did not sufficiently emphasize the amount of work and careful consideration put into preparing 66 ML-ready datasets. In the final version, we will include a discussion about the challenges of how to build ML-ready datasets.
> >
> > Briefly, all datasets included in TDC are carefully processed from the primary data resources. The raw data comes in various file formats, including machine non-readable formats, and is often inaccessible to the users. For each dataset, the raw data can be of different types, including experimental readouts, curated annotations, and metadata, and are scattered around the biorepositories and paper supplementary documents, thus requiring extensive curation to transform/link it to a format that is amenable to ML analyses. Further, many transformations and quality control steps require domain-specific expertise and familiarity with many bioinformatics and cheminformatics tools that many ML researchers do not have. Finally, each dataset in TDC comes with several dataset splits (to date, TDC implements 5 types of splits, see Figure 1) that allow for robust, fair, and reproducible model comparison and evaluation. Altogether, TDC datasets can save time and effort for ML researchers and allow the ML community to develop algorithms for key drug-related tasks.
> >
> > RE: Automatically add new datasets.
> >
> > Each task and dataset in TDC is subject to careful selection, curation, and quality control checks. Raw biomedical datasets are usually very noisy and require many preprocessing steps to make them ML-ready and biomedically meaningful. Also, each data comes from different data sources with various data formats. Thus, automatic importing of datasets is not feasible.
> >
> > RE: Future datasets contribution procedure.
> >
> > TDC is a new open-science initiative. While not discussed in the paper, we have planned and currently provide a contributor plan. The contribution guide has also been set up: https://github.com/mims-harvard/TDC/blob/master/CONTRIBUTE.md. The first TDC user group meeting will be organized in the coming months. We are actively re-structuring the TDC codebase and adding documentation for easier contribution.
> >
> > RE: Public test sets and user-reported results.
> >
> > We follow the mechanisms based on previous successes in OGB (Hu et al., 2020, NeurIPS) and WILDS (Koh et al., 2021, ICML), where we require users to explicitly provide consent to an honor code and open-source their models with fully reproducible codes. We will remove unreproducible submissions from the leaderboard. In addition, we are also planning with several institutions and pharmaceutical companies to host competitions where test sets will be hidden. Note also our response to Reviewer #1.
> >
> > RE: Sphinx API for documentation.
> >
> > TDC website in its current form clearly documents all key functionalities (https://tdcommons.ai). We note that detailed Sphinx documentation is an ongoing effort. We currently have around 70% of docstrings ready, which can be seen in the “restructure” branch: https://github.com/mims-harvard/TDC/tree/restructure.
> >
> > RE: Dataset extendability.
> >
> > TDC package supports various file formats, including, CSV, Python Pickle, SDF, etc. As TDC grows, we will continue to implement additional data loaders and utility functions for data frame libraries, such as Rapids.
> >
> > RE: Hyperparameter selection.
> >
> > The central focus of this work is the curation of datasets and tasks. We use the best hyperparameters that are optimized from the original authors as an initial benchmarking. Further improvements and model innovations would be expected from the communities. Also, note that these hyperparameters are shown to be effective across a wide range of datasets. For example, ADMET benchmarks models are tested on various MoleculeNet endpoints and DTI-DG benchmarks are tested on 7 datasets in DomainBed (Gulrajani and Lopez-Paz, 2021, ICLR).
> >
> > RE: The validity of the claim “ML SOTA models do not work well consistently for these novel realistic endpoints” in ADMET benchmarks.
> >
> > To clarify, ML SOTA methods are complicated models that report best performances in a large set of molecular property prediction tasks. The TDC benchmark shows that for several novel TDC datasets (e.g., in TDC.Caco2, RDKit has 0.393 and SOTA has 0.502), the trend discontinues, although, in the majority of the ADMET endpoints, we acknowledge SOTA methods have better performances (lines 153-155). In addition, we observe that the majority of the novel endpoints are far from being perfect, highlighting an ideal testbed for molecular property predictions.

---

> > > ### Author Response · Authors · 2021-07-09
> > > **Re: Official Blind Reviewer fHUr - Part 3**
> > >
> > > RE: Is the limited oracle setup feasible in chapter 4.3?
> > >
> > > We use docking score as an example of resource-intensive oracles (a couple of milliseconds for typical oracles such as QED versus 5 seconds for docking using vina on a CPU). This extends to computational simulation with higher fidelity (one run may take hours to days even parallelized on a supercomputer) and even the highest quality oracle-wet-lab experiments, which are even more restrictive. Thus, it is highly desirable for a model to generate high-quality results with low-resource demands.
> > >
> > > RE: What is "GIN"?
> > >
> > > GIN stands for Graph Isomorphism Network (Xu et al., 2019, ICLR). We will expand the acronym in the future version of the paper.
> > >
> > > RE: Pretraining details.
> > >
> > > Descriptions and references are provided in lines 149-152. Further, all details (including code, data, and hyperparameters) to reproduce the results are available at https://github.com/mims-harvard/TDC/tree/master/examples.
> > >
> > > RE: Randomness of the runs.
> > >
> > > For ADMET and DTI-DG benchmark, we obtain five different training and validation splits (lines 138-139 and lines 185-188). For the docking benchmark, we re-run each model with different random seeds. We will include this information in the future version.
> > >
> > > RE: Temporal splits for DTI.
> > >
> > > We have discussed the motivation and the importance of temporal splits in lines 172-174. We will expand the sections for further clarification in the future version of the paper.
> > >
> > > RE: Most TDC datasets might have been covered by DrugBank.
> > >
> > > We respectfully disagree with the reviewer that “most of the information in TDC datasets can be covered by DrugBank.” DrugBank contains properties for small-molecules while we contain compounds, gene/proteins, antibodies, antigens, peptides, MHCs, diseases, cell lines, guide RNAs, microRNAs, and chemical reactions.
> > >
> > > In fact, out of 66 datasets in TDC, there is only one dataset, TDC.DDI, for which the primary resource is DrugBank. We cite and give attribution to DrugBank when describing that dataset, which can be seen in lines 1146-1151 in the Supplementary Document.
> > >
> > > RE: Redistribution of datasets.
> > >
> > > That is an important question. We have followed the dataset and redistribution licenses from the original data providers. We have explicitly provided licenses on the website under each dataset tab and also ask the users to cite the original data source if used. We provide license information for each dataset in the Supplementary Document.

---

> > > > ### Comment · Reviewer_fHUr · 2021-07-14
> > > > **Re: Re: Official Blind Reviewer fHUr - Part 3**
> > > >
> > > > * RE: RE: Is the limited oracle setup feasible in chapter 4.3?
> > > >
> > > > Using [GPU based algorithm](https://ngc.nvidia.com/catalog/containers/hpc:autodock) can probably significantly improve the latency and throughput of the Docking methods, not to mention using distributed computation resource. This is likely the typical settings for pharmaceutical companies. So in my opinion, we should not restrict the oracle call numbers.
> > > >
> > > > * RE: RE: Most TDC datasets might have been covered by DrugBank.
> > > >
> > > > To my knowledge, DrugBank is also collecting drug related data from public resources/literature, just like what you did. It also has [BioTech drugs](https://go.drugbank.com/biotech_drugs), including peptides, antibodies, RNAs, etc. So I was thinking it is also possible that we can get a good amount of the same data on DrugBank.

---

> > > > > ### Author Response · Authors · 2021-07-14
> > > > > **RE:RE:RE Official Blind Reviewer fHUr**
> > > > >
> > > > > We thank the reviewer for additional feedback and satisfaction with most of our clarifications.
> > > > >
> > > > > If the reviewer is satisfied with additional clarifications on the main concerns, including questions about "novel" datasets and the future of the TDC ecosystem (see Parts 1, 2, and 3) and further comments below, we would greatly appreciate it if you consider updating the score. Please let us know if you have further questions or comments! Thank you.
> > > > >
> > > > > RE: How robust is the algorithm to the dataset size.
> > > > >
> > > > > We thank the reviewer for this interesting point. Note that the 22 endpoints in the ADMET benchmark (Table 2) range from data sizes of 578 to 13,130. Thus, it can directly show the robustness of the algorithm to various dataset sizes. We will add this important information as a column to Figure 2 and conduct further analysis in the next version of the paper.
> > > > >
> > > > > RE: Most TDC datasets might have been covered by DrugBank.
> > > > >
> > > > > DrugBank is highly different from TDC in the following ways.
> > > > >
> > > > > (1) DrugBank is a fantastic data repository yet its data resources do not represent ML-ready / benchmark datasets because there are no formulations of learning tasks, dataset splits, strategies for performance evaluation, etc.
> > > > >
> > > > > (2) Majority of tasks in TDC do not exist in DrugBank. Among others, DrugBank does not have CRISPR outcomes, reaction data, molecule generation oracles, quantum mechanics, paratope/epitope data, antibody developability, PPI, GDA, Peptide-MHC, antibody affinity data, and so on.
> > > > >
> > > > > (3) Even in the case of the most related ADMET properties of small-molecules, DrugBank only provides ADMET predicted features by software. In contrast, TDC provides datasets collected from more than 20 primary ADMET sources.
> > > > > We thank the reviewer for raising this point and will make sure to describe all these major differences between TDC and DrugBank in the final paper version.
> > > > >
> > > > > RE: We should not restrict the oracle call numbers.
> > > > >
> > > > > Many SOTA methods (e.g., GCPN, MolDQN) typically require oracle calls of millions. While this may be feasible via a large-scale distributed computing setup, it is costly and not scalable. In addition, our aim of this benchmark is to use docking scores as a proxy to simulate resource-intensive oracles, such as wet-lab experiments. For these oracles, for a single training run of a single oracle, you need to query millions of wet-lab experiments, which is infeasible cost-wise and time-wise.  Thus, an algorithm that can adapt quickly using only thousands of calls could be highly desirable. This shows the significance of restricting the number of oracle calls in the benchmark.

---

> > > ### Comment · Reviewer_fHUr · 2021-07-14
> > > **Re: Re: Official Blind Reviewer fHUr - Part 2**
> > >
> > > * RE: RE: Many datasets were just tabular CSV datasets that are easy to preprocess.
> > >
> > > I see. The authors should have mentioned the difficulties they encountered during the preparation of the datasets. Otherwise, people might think there are simple data crawling methods to do those from the PubMed or other database website.
> > >
> > > * RE: RE: Automatically add new datasets.
> > >
> > > Unlike typical computer vision datasets, the uncertainty in the biomedical datasets is usually very high. We should not eliminate the data that looks noisy. From the authors' descriptions above, I do agree that automatically adding/curating new datasets are not very possible, unfortunately.
> > >
> > > * RE: RE: Future datasets contribution procedure.
> > >
> > > Sounds good.
> > >
> > > * RE: RE: Public test sets and user-reported results.
> > >
> > > I personally don't like the way that OGB and WILDS are dealing with the leader-board, since it is a manual process and will not scale up. But I do look forward to hidden test sets.
> > >
> > > * RE: RE: Sphinx API for documentation.
> > >
> > > Sounds good.
> > >
> > > * RE: RE: The validity of the claim “ML SOTA models do not work well consistently for these novel realistic endpoints” in ADMET benchmarks.
> > >
> > > For example, TDC.Caco2 only has 906 data points, which is not too many. The author should also discuss the effect of the dataset size when analyzing the model performance, e.g. how robust is the algorithm to the dataset size. A training/validation curve with different training set size might be helpful for elucidating it.

---

> ### Comment · Reviewer_fHUr · 2021-07-20
> **Improve the rating.**
>
> Based on the communication, I will improve the rating from 4 to 6, given the internal complexity of the data sets and the difficulty of obtaining them. I am looking forward to the hidden test sets that the authors might have promised.

---

### Official Review · Reviewer_6PKE · 2021-07-04
**A high-quality, well-documented, extensive data availability and benchmarking resource**

**Rating:** 8
**Confidence:** 1
**Correctness:** No concerns.
**Clarity:** The paper was clear to me.

**Strengths:**

The TDC is extensive, encompassing dozens of datasets, algorithms, and benchmarks. The infrastructure includes support for public leaderboards to incentive and track development over time. Everything is tied nicely together by a Python package that makes TDC accessible.

The included cases studies are useful to demonstrate the capabilities of the TDC.

**Weaknesses:**

Although the authors are not responsible for producing any of the datasets, only curating them, I would have appreciated some discussion of the ethics of collection of these datasets, i.e., were they collected with patient consent? Are there any latent HIPAA issues?

**Additional Feedback:**

Thank you for the response and clarifications.

**Documentation:**

The TDC website is slick and extensive. There are clearly a lot of resources going into this effort, so I have no concerns about longevity or maintenance. Everything is publicly available.

**Ethics:**

No concerns.

**Relation To Prior Work:**

The TDC seems go far beyond existing resources in this domain.

**Summary And Contributions:**

The paper presents an overview of the Therapeutic Data Commons, a resource for datasets, algorithms, benchmarks, and leaderboards that all address machine learning problems related to drug discovery.

---

> ### Author Response · Authors · 2021-07-09
> **Re: Official Blind Reviewer 6PKE**
>
> We thank the reviewer for the feedback and noting that TDC is a high-quality, well-documented, and extensive data collection and benchmarking resource.
>
> RE: The ethics of collection of these datasets.
>
> TDC currently does not involve human subject research data. Current data types in TDC originate from biological and scientific research, including compounds, gene/proteins, antibodies, antigens, peptides, MHCs, diseases, cell lines and model organisms, guide RNAs, microRNAs, and chemical reactions. We thank the reviewer for their suggestion to develop a procedure for ethics review before adding new datasets, e.g., checks that anyways need to be done by an IRB when data involves human subjects research. TDC is actively maintained and developed by a team of volunteers and community members. As we continue to grow TDC, we will include guidelines, implement best practices (e.g., Wiens et al., Nature Medicine, 2019) and provide checklists (such checklists exist, for example, for clinical ML (Norgeot et al., Nature Medicine 2020) but not yet for therapeutics ML) to ensure that human subject research data (when included in TDC) are ethically sourced.

---

### Official Review · Reviewer_uEa5 · 2021-07-06
**An extensive compilation of datasets, models, and benchmarks with a potential to accelerate machine learning for drug development**

**Rating:** 9
**Confidence:** 4

**Strengths:**

The main strength is that the platform includes a diverse collection of machine learning tasks and accompanying datasets which simultaneously are of interest to drug development and machine learning communities. The platform also standardizes evaluation on the tasks through leaderboards and implementations of data processing and evaluation methods. Experiments with state-of-the-art methods are reported and areas for new algorithm development are highlighted.

**Weaknesses:**

The main weakness is the lack of discussion on (1) provisions to reduce overfitting of approaches to the chosen datasets and tasks (e.g. are the test sets hidden); (2) how can the submitted test results be independently validated by the platform; and (3) whether the platform will collate information on well-performing machine learning models, e.g. by enabling users to share model checkpoints or code.

**Additional Feedback:**

I am intrigued by the outlined research (line 66 in main paper) that the platform can help advance, e.g. data efficient learning and robust learning against distribution shifts for multi-modal data. I am excited about the developments to be enabled by the platform.

Minor questions

Can you make provisions for domain scientists to share guidelines on acceptable performance numbers to be met by the machine learning products on the benchmarks that will support taking them to the laboratories for further testing?

What is the rationale for not supporting model sharing or easy integration of existing model libraries, e.g. OpenChem [72], with the benchmarks?

The appendix mentions that “TDC does not involve human subjects research”. However, one proposed uses of TDC, mentioned in the main paper (line 66), is causal inference to quantify “treatment response across patient groups”. I am confused by the two statements. Is the claim that future research on TDC does not involve human subject whereas included datasets might have?

**Clarity:**

The authors meticulously detail each dataset and task; and their context in the drug development pipeline. The paper is quite well written. The experiment setup, baselines, and results are summarized effectively.

**Correctness:**

Details of datasets curation and motivating applications for each are given. Evaluation metrics for the different machine learning tasks seem to be comprehensive. Experiment design follows recommended machine learning practices, e.g. reporting cross validation performance.

**Documentation:**

Dataset details and their intended uses are documented. Data and codes for benchmarking are publicly available. Codes for reproducing experiments is also given.

**Ethics:**

Ethics implications from the currently shared datasets and tasks are adequately addressed. I would suggest building in a procedure for doing ethics review before adding the new datasets, e.g. checks that will anyways be done by an IRB for data involving human subjects research.

**Relation To Prior Work:**

The platform differentiates from prior work by focusing on tasks related to drug development, and providing datasets, benchmarks, and accompanying code for evaluation.

**Summary And Contributions:**

The work presents a platform for supporting machine learning research on therapeutics. It consists of easily accessible datasets (66 in total) for a variety of machine learning tasks (instances of graph representation learning, domain generalization, and generative modeling), and support tools for model building and evaluation. The authors curate a diverse set of tasks in context of drug development which challenges the community to tackle cutting edge machine learning problems. Progress on algorithms on these tasks has the potential to enable scientific discovery in this high-impact domain. A comprehensive suite of benchmarks is developed to measure progress on the tasks along with code for easily performing the evaluation. With a well-documented set of support tools, the platform has the potential for wide adoption by the machine learning and domain scientists working on drug development, and thus, may accelerate progress in the field.

----
After author response

Thanks for responding in detail on unseen test sets and independent evaluation. I would encourage authors to collate references to laboratory studies for the tasks to give users an idea of acceptable performance on benchmarks.

---

> ### Author Response · Authors · 2021-07-09
> **Re: Official Blind Reviewer uEa5**
>
> We thank the reviewer for the positive feedback and constructive advice. The reviewer has raised great points, and below is our response to their comments.
>
> RE: Provisions to reduce overfitting to the datasets and tasks.
>
> We agree with the reviewer that overfitting is possible when test sets are public. However, most TDC datasets specify ML tasks that are far from being solved, meaning that improvement in performance could still suggest meaningful algorithm innovation. Further, whenever possible, we provide multiple datasets for the same ML task. For example, we provide 22 datasets for the ADMET task. These datasets are diverse (e.g., generated by different biotechnological platforms, measured in different tissues, etc) and form “Benchmark Groups” (see section 4.1 in the paper). This unique grouping of individual benchmarks into benchmark groups mitigates the issue of overfitting to some extent as a well-performing model needs to perform well across multiple datasets.
>
> Finally, benchmarks in TDC follow the structure of prominent benchmark projects, e.g., OGB (Hu et al., 2020, NeurIPS), WILDS (Koh et al., 2021, ICML), TAPE (Rao et al., 2019, NeurIPS), MLPerf (Mattson et al., 2020, MLSys), MoleculeNet (Wu et al., 2018, Chemical Science), GuacaMol (Brown et al., 2019, JCIM) that also have public test sets (or oracles). Beyond that, we are in the process of setting up collaborations with several institutions and pharmaceutical companies where we will host competitions with hidden test sets on the TDC. To preserve the integrity of test results, we will not release the test set to the public.
>
> RE: Independent validation of submitted test results.
>
> We follow the mechanisms based on previous successes in OGB (Hu et al., 2020, NeurIPS) and WILDS (Koh et al., 2021, ICML) and require users to explicitly agree to the honor code and open-source their models with fully reproducible codes (https://bit.ly/3wt6mLt). We remove irreproducible submissions from the leaderboard.
>
> RE: Collate information on well-performing machine learning models.
>
> That is a great point. For each leaderboard, TDC provides a ranked list of submitted models where each submission has a pointer to a public implementation of the model and its relevant publication. For example, confer the leaderboard for molecule generation task: https://tdcommons.ai/benchmark/docking_group/drd3.
>
> RE: What are acceptable performance numbers for further laboratory testing?
>
> That is a great question for which there are no easy or universal answers. For most tasks, there is no consensus on the minimum required performance of the model in order to transition into biomedical implementation. This is because laboratory testing can be complicated; beyond quantitative measures of performance, in many cases, measures of the cost of deployment or impact (e.g., on financial outcomes) might also be considered (Wiens et al., Nature Medicine, 2019). To the best of our knowledge, we are not aware of any standards specifying minimal required performance for downstream laboratory validation. The reviewer has raised an important point, and, for future datasets, we plan to include pointers to published studies that carried out laboratory testing--such studies could provide guidelines for acceptable performance numbers.
>
> RE: Supporting integration of existing model libraries with the benchmarks.
>
> TDC is focused on providing datasets, task definitions, and supporting functions (e.g., data processors and loaders, model evaluators) to evaluate and compare models that are developed by users. That said, we provide easy-to-use wrappers that connect TDC resources with popular model libraries, including DeepPurpose (Huang et al., 2020, Bioinformatics) on the ADMET benchmark: https://github.com/mims-harvard/TDC/tree/master/examples/single_pred/admet.
>
> RE: Involvement of human subjects research.
>
> TDC currently does not involve human subject research data. Current data types in TDC originate from biological and scientific research, including compounds, gene/proteins, antibodies, antigens, peptides, MHCs, diseases, cell lines and model organisms, guide RNAs, microRNAs, and chemical reactions. We thank the reviewer for their suggestion to develop a procedure for ethics review before adding new datasets, e.g., checks that anyways need to be done by an IRB when data involves human subjects research. TDC is actively maintained and developed by a team of volunteers and community members. As we continue to grow TDC, we will include guidelines and implement best practices (e.g., Wiens et al., Nature Medicine, 2019) to ensure that human subject research data (when included in TDC) are ethically sourced.

---

### Decision · Program_Chairs · 2021-07-26

**Decision:**

Accept

**Comment:**

The paper makes a significant contribution in terms of datasets, models, and public infrastructure that has the potential to accelerate machine learning for drug discovery.